# Quality of Life and Associated Factors among Cancer Patients Receiving Chemotherapy during the COVID-19 Pandemic in Thailand

**DOI:** 10.3390/ijerph21030317

**Published:** 2024-03-09

**Authors:** Porawan Witwaranukool, Ratchadapa Seedadard, Suphanna Krongthaeo, Yosapon Leaungsomnapa

**Affiliations:** 1Ramathibodi School of Nursing, Faculty of Medicine, Ramathibodi Hospital, Mahidol University, Bangkok 10400, Thailand; suphanna.kro@mahidol.edu; 2National Cancer Institute, Bangkok 10400, Thailand; ratchadapa.see@gmail.com; 3Phrapokklao Nursing College, Faculty of Nursing, Praboromarajchanok Institute, Chanthaburi 22000, Thailand; yosapon@pnc.ac.th

**Keywords:** quality of life, cancer, chemotherapy, COVID-19, Thailand

## Abstract

The dynamics of the COVID-19 pandemic have significantly changed since its initial outbreak. This study aimed to investigate the quality of life (QoL) of patients with cancer receiving chemotherapy in the specific context of Thailand during the COVID-19 pandemic. A cross-sectional study was conducted with 415 patients with cancer. Instruments used were a demographic and clinical characteristics form, the Edmonton Symptom Assessment Scale (cancer symptom burden), Strategies Used by People to Promote Health (self-care self-efficacy), and a Thai version of the Brief Form of the WHO Quality of Life Assessment. Data were analyzed using descriptive and inferential statistics. The participants had an average age of 56 years. They reported a moderate level of QoL across all domains and for the overall QoL during the pandemic. The results of the multiple linear regression model indicated that positive self-care self-efficacy, being married, having health insurance, stage of chemotherapy, and reduced cancer symptom burden were significant predictors of overall QoL (adjusted R^2^ = 0.4940). Positive self-care self-efficacy also emerged as a primary predictor, positively influencing all QoL domains and overall QoL (*p* < 0.001). These findings emphasize the significance of self-care self-efficacy in enhancing the QoL of patients with cancer undergoing chemotherapy during the pandemic. Integrating interventions to bolster self-care self-efficacy into the care plans for these patients can help them manage their symptoms, cope with the side effects of cancer treatment, and enhance their overall well-being.

## 1. Introduction

Cancer is a prominent global contributor to mortality, accounting for nearly one in every six deaths, based on worldwide estimates [1]. In 2022, it was also estimated that globally around 20 million new cases of cancer emerged, resulting in approximately 9.7 million deaths [2]. In Thailand, 381 people are diagnosed with cancer each day, resulting in approximately 130,000 new cases annually [3]. Furthermore, cancer causes the death of 230 Thai people daily, totaling 84,000 deaths per year [3]. While medical advancements have expanded the range of treatment options available to patients with cancer, chemotherapy remains a crucial and frequently employed therapeutic modality [4]. However, the global outbreak of the COVID-19 pandemic has had profound implications on various aspects of healthcare delivery including the provision of chemotherapy to cancer patients [5]. Healthcare professionals were under pressure to continue ensuring that the clinical needs of patients with cancer were still met during this time [5]. Additionally, such patients were faced with heightened vulnerability due to their compromised immune systems, necessitating strict infection control measures and, in some cases, treatment delays or modifications [6]. In Thailand, the situation regarding COVID-19 has evolved since the pandemic’s initial onset [7]. The Thai nation has successfully implemented a range of effective strategies in its battle to contain the virus. The strict infection control measures have allowed the nation to gradually return to normal operations while still prioritizing safety through the continued use of protective measures such as masks and social distancing [8]. During the fifth wave of the pandemic, the Thai government has implemented measures that are relatively more lenient compared to those in the previous waves [7]. However, the prolonged enforcement of these policies had the potential to negatively impact the overall quality of life (QoL).

The impact of cancer and chemotherapy on patients’ QoL spans various dimensions, including physical, psychological, social, and financial aspects, often leading to significant distress, fatigue, and diminished ability to perform daily activities [9]. The COVID-19 pandemic has compounded these challenges, disrupting cancer care and treatment schedules. Previous studies conducted during the COVID-19 pandemic have highlighted the detrimental effects of the pandemic on the QoL, both physically and mentally, of survivors of breast cancer [10] and head and neck cancer [11]. However, staying at home and spending more time with family members had a positive impact on their well-being [10]. Moreover, patients diagnosed with pancreatic cancer faced emotional distress and concerns related to the potential impacts of COVID-19 [12]. This has led to treatment delays, ultimately resulting in the transition of previously curable tumors to incurable ones [12]. Additionally, individuals undergoing chemotherapy have reported increased symptom burdens, encompassing both physical and psychological symptoms, and a diminished overall QoL [13]. Earlier research has shown that patients with advanced cancer often bear a heavy symptom burden, affecting their ability to work, enjoy life, engage in daily activities, and maintain their physical health [14]. In the case of patients with colorectal cancer receiving chemotherapy, a high symptom burden has also been linked to a reduced QoL [15]. It is important to highlight that self-efficacy in self-care management significantly influences the QoL among patients with cancer [16,17]. The concept of self-efficacy highlights the link between strong confidence in performing tasks and the likelihood of achieving desired outcomes, including maintaining QoL [18]. Within the context of cancer care, self-care management encompasses several key activities: medication and drug management, lifestyle adjustments, psychological well-being, social network support, accessing knowledge and information, navigating healthcare systems, and medical decision-making [19]. Patients with cancer encounter challenges related to their treatment, vulnerabilities of their immune system, and the additional risks posed by COVID-19 [12]. Consequently, those with greater ability and confidence in managing self-care activities may experience improved emotional and physical well-being during the pandemic. While the aforementioned literature provides insights into the QoL of various cancer types during the global COVID-19 pandemic, there is a lack of research examining additional factors that may influence the QoL of patients with cancer during this period.

In Thailand, the COVID-19 pandemic has impacted how individuals experience and manage illnesses, including cancer. The distinctive impact of the COVID-19 pandemic on Thai patients with cancer, highlights the role of Buddhist philosophy in maintaining a positive outlook [20], despite the challenges of isolation and economic stress that have strained traditional family support systems. This cultural context is essential for understanding the specific obstacles faced by the Thai healthcare system during the pandemic. Thai cancer patients’ management of their health challenges their ability to access healthcare services, and the level of community support they receive may substantially differ from practices in other regions. The Thai healthcare system, including cancer services, offers a mix of public and private healthcare services, where the accessibility and quality of cancer care can vary significantly, often influenced by socioeconomic status and geographic location [3]. Community support, deeply rooted in Thai culture, plays a critical role in the health management of patients with cancer, providing emotional, social, and sometimes financial assistance, thereby enhancing the coping mechanisms of patients navigating their treatment journey. Furthermore, traditional Thai health beliefs and practices may also influence the approach to cancer care, with some patients seeking alternative or complementary treatments in conjunction with conventional medical therapies [18]. This may lead patients to incorporate traditional medicine, meditation, and spiritual practices alongside conventional cancer treatments. The differences in the health management of Thai patients with cancer are crucial in understanding and developing tailored approaches to improve their QoL, especially during challenging times like the COVID-19 situation.

Considering all these elements, it is essential to conduct a study in Thailand that aimed to investigate changes in the QoL of patients with cancer before and during the COVID-19 pandemic and to explore factors influencing their QoL while undergoing chemotherapy.

## 2. Materials and Methods

### 2.1. Study Design

This study adopted a descriptive, cross-sectional design and employed a convenient sampling method for data collection.

### 2.2. Setting

The National Cancer Institute in Thailand was chosen as the primary data collection site. This institute is renowned as one of the leading comprehensive cancer treatment centers in Asia. It serves as a cancer care hub with six branches strategically located across various geographical areas of Thailand [21]. Certain cancer patients who cannot be treated at local branches are referred for advanced treatment at the National Cancer Institute. The central mission of this institute is to provide advanced cancer treatments to individuals diagnosed with cancer across all provinces, offer training programs for oncological healthcare professionals, and conduct cutting-edge research in the field of oncology [21]. Given its significant role, this setting not only caters to a large number of patients with cancer but also plays a pivotal role in providing healthcare professionals with education and opportunities to be involved in cancer research.

### 2.3. Participants

This study recruited patients with cancer undergoing active chemotherapy or follow-up chemotherapy programs at the Thai National Cancer Institute. Eligible participants were individuals who had undergone at least one chemotherapy treatment in either an inpatient or outpatient care unit; were over 18 years old; and expressed a willingness to take part in the survey. Criteria applied to exclude cancer patients from the study were patients who had undergone other cancer treatment in the period between the previous chemotherapy session and the time of the data collection; were unable to read and answer the survey questions on paper; had psychological issues; and expressed unwillingness to participate in the survey.

The sample size for this study was determined using the G*Power program. The significance level was set at 0.05, with a test power of 95%, and the effect size was 0.2 [22]. The final sample comprised 319 patients with cancer. Considering any possible attrition stemming from incomplete questionnaires or patients dropping out of the study, the required sample size was increased by 30% [23]. Therefore, the target population for this study was 415 patients with cancer.

### 2.4. Procedure

The study was carried out between January and June 2023, with approval numbers COA. MURA2023/9 from the Ethics Committee of the Faculty of Medicine Ramathibodi Hospital and COA. 010/2023 from the National Cancer Institute, Thailand. The study adhered to the guidelines outlined in the Declaration of Helsinki. Potential participants received an information sheet and any queries or concerns regarding the study were addressed before they made the decision to participate. Those who agreed to participate signed a consent form before completing the questionnaire, which required approximately 10 min to fill out. Participants completed the questionnaire while waiting for their appointments with oncologists. The questionnaires were returned in sealed envelopes to ensure anonymity and confidentiality. Throughout the data collection process, strict adherence to the COVID-19 guidelines set forth by the Thai Ministry of Public Health was followed. Measures included the provision of waterless hand gel; the allocation of private rooms (one room per person) for participants to complete the questionnaire; mandatory mask-wearing within the cancer institute premises; and ensuring the maintenance of social distancing guidelines [24].

### 2.5. Measures

This study utilized four measures to gather data: a demographic and clinical characteristics form, the Edmonton Symptom Assessment Scale (ESAS), the Strategies Used by People to Promote Health (SUPPH), and a Thai version of the Brief Form of the WHO Quality of Life Assessment (WHOQOL-BREF-THAI). The latter three measures have been successfully applied in several countries, including Thailand, and exhibit good psychometric properties. Their validity and reliability, after being translated into Thai, are considered satisfactory for use with various types of cancer.

The demographic characteristics, which encompassed age, gender, marital status, education level, occupation, the sufficiency of income during the COVID-19 pandemic, and insurance status, were collected through direct inquiries with the participants. Clinical characteristics, such as the type and stage of cancer, the history of prior cancer treatments, and the sequence of chemotherapy administered during COVID-19, were gathered from patients’ medical records after obtaining authorization from the National Cancer Institute to access these documents. 

#### 2.5.1. Edmonton Symptom Assessment Scale (ESAS)—Cancer Symptom Burden

The Thai version of ESAS has been demonstrated to be a valid and reliable tool for evaluating the severity of symptoms in cancer patients [25]. In this study, we utilized this instrument to assess the symptom burden experienced by cancer patients following chemotherapy. The Thai version of ESAS consists of 10 items categorized into two domains: (1) physical domain (items 1–6), containing pain, dyspnea, appetite, nausea, fatigue, and drowsiness and (2) psychological domain (items 7–9), containing anxiety, depression, and other well-being. Additionally, item 10 assesses other symptoms that cancer patients may experience. Each item is rated on a numerical scale ranging from “No symptom” (0) to “Highest severity of symptom” (10). In this study, item 10 was not experienced by the potential participants, and therefore, 9 items were analyzed with total possible scores ranging from 0 to 90. Higher scores indicate a greater perception of the burden of cancer-related symptoms. The Cronbach’s α coefficient for ESAS in this study was calculated as 0.75, indicating a good level of internal consistency.

#### 2.5.2. Strategies Used People to Promote Health (SUPPH)—Self-Care Self-Efficacy

SUPPH questionnaire was developed to assess the self-confidence of cancer patients in their ability to manage self-care tasks while undergoing treatment and it has demonstrated strong psychometric properties [26]. In the context of this study, SUPPH was used to assess cancer patients’ confidence in executing self-care strategies during the COVID-19 pandemic. The questionnaire encompasses 29 items, which are divided into three distinct subscales: stress reduction (items 1–10), decision-making (items 11–13), and positive attitude (items 14–29). Respondents are asked to rate each item on a 5-point Likert scale, ranging from “Very little” (1) to “Quite a lot” (5). The possible scores range from 29 to 145, with higher scores indicating a stronger belief in one’s ability to engage in self-care effectively. In this study, the Thai version of SUPPH was available in Thailand. The author who translated the SUPPH questionnaire into Thai adhered to the established translation guidelines, which was followed by a comprehensive assessment of each item’s content validity by a panel of experts [27]. These steps were then undertaken before utilizing the Thai version of SUPPH to assess the self-care self-efficacy of women with breast cancer in her study [28]. The internal consistency of the total SUPPH in this study was assessed using Cronbach’s α coefficient, resulting in a value of 0.88, which indicates higher than acceptable internal consistency.

#### 2.5.3. The Thai Version of the Brief Form of the WHO Quality of Life Assessment (WHOQOL-BREF-THAI)—Quality of Life

The Thai version of WHOQOL BREF-THAI has undergone validation for assessing the QoL in Thai populations dealing with chronic conditions, including individuals diagnosed with cancer [28]. This version has been modified from the WHOQOL-100 to ensure cultural relevance and appropriateness for the Thai demographic. Due to its user-friendly format and its focused aspects concerning cancer patients’ QoL during the COVID-19 pandemic, WHOQOL BREF-THAI serves as a key dependent variable in our study. This instrument comprises 26 items, and it is employed to evaluate patients’ perceptions of their QoL. These items are divided into four domains: physical health, psychological health, social relationships, and environment. Respondents are asked to rate each item using a 5-point Likert scale, ranging from “Not at all” (1) to “An extreme amount” (5). For assessing QoL levels, the possible scores for each domain and the overall QoL are as follows: the physical health domain scores range from 7 to 35; the psychological health domain scores range from 6 to 30; the social relationships domain scores range from 3 to 15; the environment domain scores range from 8 to 40; and the overall QoL scores range from 26 to 130. To categorize QoL levels within each domain, cut-off points from the previous Thai version of WHOQOL BREF-THAI were applied. In the physical health domain, total scores were split into poor (7–16 points); moderate (17–26 points); and good (27–35 points). Psychological health scores were categorized into poor (6–14 points); moderate (15–22 points); and good (23–30 points). Social relationships scores were divided into poor (3–7 points); moderate (8–11 points); and good (12–15 points). In the environment domain, scores were categorized as poor (8–18 points), moderate (19–29 points), and good (30–40 points). The overall QoL scores were grouped as poor (26–60 points), moderate (61–95 points), and good (96–130 points). In our study, we calculated the internal consistency of the WHOQOL BREF-THAI’s subdomains and overall scale. The Cronbach’s α coefficients were found to be 0.83 for physical health, 0.87 for psychological health, 0.84 for social relationships, 0.85 for environment, and 0.85 for the total scale. These results indicate that the WHOQOL BREF-THAI demonstrates a high level of internal consistency and reliability.

### 2.6. Statistical Analysis

The data analysis was conducted using the STATA 18 software. Descriptive statistics were employed to examine socio-demographic factors, clinical characteristics, cancer symptom burden, self-care self-efficacy, and QoL. To assess differences in QoL values before and during the COVID-19 pandemic, we compared the mean scores obtained from the WHOQOL BREF-THAI instrument. For pre-COVID-19 data, we referred to a study by Mechalearn et al. [29], and we had authorization to use their data. This study explored QoL among cancer patients and their caregivers during chemotherapy treatment before the pandemic. The reason to use Mechalearn et al.’s study is that their participant characteristics were comparable to those in our study. Moreover, both Mechalearn et al.’s study and ours utilized the WHOQOL-BREF-THAI questionnaire to measure QoL in cancer patients, making their study a suitable reference for exploring the differences in QoL scores before and during the COVID-19 pandemic. The mean scores of WHOQOL-BREF-THAI were also calculated by utilizing Hedges’ g effect size an effect size of ≥0.8 was considered indicative of a large effect [30]. In our study, we explored the relationships between socio-demographic factors, clinical characteristics, cancer symptom burden, and self-care self-efficacy with various QoL domains. To analyze these relationships, we applied a range of statistical methods, including T-tests, One-way ANOVA, and Pearson’s correlation analysis. To identify predictive factors for QoL domains, we employed a stepwise multiple linear regression analysis. A significance level of *p* < 0.05 was used to determine statistical significance.

## 3. Results

### 3.1. Sample Characteristics

Table 1 presents the characteristics of 415 cancer patients who took part in a survey. The participants’ average age was 56 ± 11 years. The majority of participants were female (75.4%), married (60.7%), and had completed primary school (37.3%). A significant portion of the participants were retired or housewives (39.3%) and experienced income insufficiency during the COVID-19 pandemic (55.4%). Regarding health coverage, over half (58.55%) of cancer treatment expenses were covered by universal health coverage. In terms of clinical characteristics, among the various cancer types, breast cancer had the highest prevalence (34.2%), followed by gastrointestinal cancer (29.2%) and gynecological cancer (22.7%). A proportion of cancer cases were categorized as stage 4 with metastasis (40%), while an additional 34.5% were classified as stage 3. In the sequence of chemotherapy during the COVID-19 pandemic nearly half of the participants received fewer than four cycles of chemotherapy (56.63%). As for the ESAS score, assessing cancer symptom burden, an average total score of 12.16 ± 2.34 was recorded, with a maximum possible score of 90. In addition to the SUPPH score, measuring participants’ self-care self-efficacy, the average total score was 103.14 ± 21.71, with a potential maximum score of 145.

### 3.2. QoL Outcomes

Table 2 presents a comparative analysis of the QoL before the onset of the pandemic, utilizing data from the previous study (as the control cohort) [24], and the QoL during the COVID-19 pandemic, utilizing data from the current study (as the current cohort). In general, QoL levels across all domains and for the overall QoL in the current cohort were rated as moderate. In terms of physical health, the mean score for the current cohort was higher (mean = 22.84) than that of the control cohort (mean = 22). Similarly, the psychological health mean score exhibited a higher value in the current cohort (mean = 21.63) compared to the control cohort (mean = 16.54). However, contrasting these findings, the current cohort displayed slightly diminished scores in the domains of social relationships and environment. Specifically, the mean score for social relationships was lower in the current cohort (mean = 9.81) compared to the control cohort (mean = 11.54). Likewise, the environment domain showed a reduced mean score in the current cohort (mean = 27.75) in comparison to the control cohort (mean = 32.24). Furthermore, evaluating the overall QoL during the COVID-19 pandemic it was observed that the mean score for the current cohort (mean = 87.14) was lower than that of the control cohort (mean = 90.38). These findings suggest that while certain aspects of health experienced positive changes, the overall impact of the pandemic on QoL was adverse. To statistically analyze the differences in the mean scores for all QoL domains and the overall QoL of the two cohorts, an independent-sample t-test was conducted. The results of this analysis indicated a statistically significant difference in all domains of QoL as well as the overall QoL (*p* < 0.05 for all domains and for the overall QoL).

### 3.3. Factors Affecting QOL

Table 3 reports the comparisons of the baseline characteristics, cancer symptom burden (ESAS), and self-care self-efficacy (SUPPH) with the domains of QoL as well as the overall QoL.

The age of the participants was negatively associated with the domains of physical health (r = −0.125; *p* < 0.05) and social relationships (r = −0.155; *p* < 0.001). Marital status displayed statistically significant differences across all QoL domains (*p* < 0.05 for all domains and the overall QoL), except for the psychological domain. Similarly, participants with a university degree or higher exhibited elevated QoL scores across all domains, with statistical significance in the education level (*p* < 0.05 for all domains and the overall QoL). Adequate income during the COVID-19 pandemic demonstrated an association with improved QoL in the domains of physical health (*p* = 0.001), psychological health (*p* = 0.038), and social relationships (*p* = 0.026). Furthermore, participants covered by government enterprise employee health insurance exhibited higher QoL scores in the domains of social relationships (*p* = 0.005) and environment (*p* = 0.023), as well as in the overall QoL (*p* = 0.017), compared to those under universal health coverage. The sequence of chemotherapy cycles yielded statistically significant disparities across all domains (*p* < 0.05 for all domains and the overall QoL), with the exception of social relationships. Furthermore, a negative correlation was observed between ESAS and the physical health domain (r = −0.112; *p* < 0.05), indicating a tendency for higher cancer symptom burden in cases of poorer physical health. The participants also revealed a significant positive association between SUPPS and all domains of QoL and the overall QoL (*p* < 0.001 for all domains and the overall QoL). This suggests that the participants with greater self-care self-efficacy reported higher QoL.

### 3.4. Predictors of QOL 

Table 4 presents an outline of the predictive factors exerting influence on QoL. Those factors that exhibited a statistically significant correlation with the domains of QoL were selected to formulate models for each respective domain. Notably, the physical health model demonstrated associations with SUPPS (*p* < 0.001), ESAS (*p* < 0.001), marital status (*p* = 0.003), sequence of chemotherapy (*p* = 0.006), and adequate income (*p* = 0.20). In parallel, the psychological health model manifested connections with SUPPS (*p* < 0.001), ESAS (*p* = 0.002), sequence of chemotherapy (*p* = 0.015), and health insurance (*p* = 0.041). Similarly, the social relationship model displayed associations with SUPPS (*p* < 0.001), health insurance (*p* = 0.007), and age (*p* = 0.017). Likewise, the environment model demonstrated correlations with SUPPS (*p* < 0.001), health insurance (*p* = 0.003), and sequence of chemotherapy (*p* = 0.046). The overall QoL model exhibited significant associations with SUPPS (*p* < 0.001), marital status (*p* =0.008), ESAS (*p* = 0.011), health insurance (*p* = 0.018), and sequence of chemotherapy (*p* = 0.047). Considering the collective influence of these factors, the overall QoL model explicated 49.40% of the variance. It is of significance to underscore that SUPPS emerged as the primary predictor across all QoL models.

## 4. Discussion

This study aimed to assess QoL and its related factors in patients with cancer undergoing chemotherapy during the COVID-19 pandemic in Thailand. Our study revealed that all domains of QoL, encompassing physical health, psychological health, social relationship, and environment, were found to be at moderate levels on the WHOQOL BREF-THAI measurement. Our results indicated that patients with cancer were experiencing positive outcomes in terms of both physical and psychological well-being, leading to a heightened sense of comfort. These findings align with a previous Western study that employed similar measurements and studied the same population during the COVID-19 pandemic [31]. In Thailand, the healthcare system’s adaptation to the COVID-19 pandemic has been crucial in preserving QoL for patients with cancer undergoing chemotherapy. The dedication of healthcare professionals to the effective management of chemotherapy side effects, along with the integration of home care services that augment familial support, has been key in maintaining patients’ physical well-being [32]. This may contribute significantly to the overall effectiveness of the healthcare system in supporting patients with cancer.

We observed significant differences in all domains of QoL in cancer patients between the periods before and during the COVID-19 pandemic. When comparing the reference values of the control cohort in a pre-COVID-19 pandemic using WHOQOL-BREF-THAI [29], it was observed that during the COVID-19 pandemic, the scores for physical health and psychological health had increased, while the scores for social relationships and the environment had decreased. These findings align with the study by Ciążyńska et al. [33] reporting lower scores of global health status (QoL scale) and social functioning as measured by EORTC QLQ-C30 among patients with cancer during the COVID-19 condition compared to normal conditions. Our findings may be associated with the protective behavior guidelines implemented in Thailand during the COVID-19 pandemic, such as social distancing, mask-wearing, and restrictions on large gatherings [24]. These behaviors may have limited cancer patients’ ability to engage in regular social interactions, resulting in a decline in the quality and quantity of social relationships. Furthermore, the COVID-19 situation in Thailand has also led to disruptions or limitations in support services. This includes peer support and home care assistance for cancer patients [34]. The persistence of disruptions during the fifth wave of the COVID-19 pandemic in Thailand likely continued to impact the level of social support available, leading to a diminished supportive environment for cancer patients. This ongoing situation may have contributed to the reduced availability of social resources and assistance, further challenging the well-being of these patients. Our study also revealed that the physical health and psychological health domains of QoL showed higher scores during the COVID-19 pandemic compared to the reference cohort [29] which contradicts the study by Ciążyńska et al. [33]. The possible reason for our results is that, in Thailand, healthcare facilities and healthcare professionals may have been more attentive to cancer patients during the COVID-19 pandemic, offering specialized care and telemedicine services to ensure their well-being amid the added risks posed by COVID-19 [35]. This focused attention and support from healthcare professionals could have positively influenced the patients’ perception of their physical and psychological health [35]. Our results also show the scores of cancer symptom burden (ESAS), including physical and psychological symptoms that may be related to our results that these symptoms were under 50% of the possible scores. This suggests that cancer patients who perceive fewer physical and psychological symptoms may experience a better overall sense of well-being, even if some symptoms are still present but not severe.

Many factors are associated with the QoL of cancer patients during the COVID-19 pandemic. Notably, cancer patients with a higher education level, including those who had graduated from university or attained higher education, reported higher scores in all domains and for the overall QoL. However, these results are partially similar to the study conducted by Nguyen et al. [36], which investigated the QoL related to health by using the EORTC QLQ-C30 questionnaire among patients with cancer during the COVID-19 pandemic. The authors noted that, among cancer patients, a higher level of education was only specifically linked to improvements in the emotional functioning domain of QoL. The possible reasons for our results are that higher education among cancer patients may lead to better understanding and adherence to health-related practices, including cancer treatment protocols. Improved comprehension and compliance with medical recommendations can contribute to a reduction in the side effects of cancer treatment, consequently positively impacting their overall health and well-being [37]. Additional findings from our study indicate that married cancer patients demonstrated higher scores in the physical health, social relationships, and environmental domains, as well as the overall QoL, compared to those who were not married. The observed higher scores in these domains among married cancer patients may be attributed to the mutual support and encouragement they provide for each other [38].

Regarding income sufficiency, our study revealed a significant finding with approximately 50% of cancer patients reporting no income during the COVID-19 pandemic. This lack of income was found to be associated with lower scores in physical health and psychological health, as well as social relationships domains of QoL. These results partially align with the findings of previous studies conducted during the COVID-19 pandemic in terms of the association between income and social relationship domains [36,39]. In Thailand, the COVID-19 situation has significantly disrupted social networks and financial support systems, resulting in negative income effects [40]. Previous studies highlighted a concerning trend of increasing perceived income burden among cancer patients which correlated with rising expenditures of cancer treatment [39,41]. In Thailand, these expenditures are covered by the universal health coverage scheme which provides basic medical treatment for Thai citizens. However, the findings of our study indicated that cancer patients enrolled in the universal health coverage scheme tended to report lower QoL scores in social and environmental domains compared to those with government enterprise employee health insurance, a more specialized insurance plan tailored for government employees and their families. This discrepancy can be attributed to the fact that the universal health coverage scheme falls short of offering comprehensive coverage for essential cancer treatments and medications. Consequently, cancer patients covered under the universal health coverage scheme may face financial challenges that have a detrimental impact on their overall QoL.

Further findings from this study revealed an association between the frequency of chemotherapy cycles and QoL. Cancer patients who received a higher number of chemotherapy cycles showed higher scores in physical health, psychological health, and environmental domains, as well as overall QoL. These results contrast with an earlier study conducted during the COVID-19 pandemic in Vietnam, which did not find a significant association between chemotherapy cycles and all QoL domains [36]. However, our findings align with an earlier pre-pandemic study that focused on administering chemotherapy to breast cancer patients, supporting the notion that increased chemotherapy cycles could positively impact the QoL [42]. One potential explanation for our results is the adaption of oncological healthcare professionals in Thailand during the COVID-19 pandemic. They have provided home-based chemotherapy to cancer patients to ensure the continuity of their treatment and to improve cancer patients’ QoL [6]. By adhering to their treatment schedule, patients may have perceived that their disease is being actively managed, leading to a positive impact on their overall well-being. Furthermore, our study identified a negative association between age and QoL in cancer patients. This aligns with the results of a previous study, which indicated that higher age was associated with a decline in the physical composite summary-related QoL among cancer patients, both before and during the COVID-19 pandemic lockdown [43].

When considering the burden of cancer symptoms with QoL, we found that the cancer patients who had higher cancer burden symptoms were more likely to have poorer physical health domain of QoL than those who were not. A previous study has established a clear negative correlation between the symptom burden of cancer patients and their physical function [44]. This symptom burden was also directly linked to cancer treatment. This implies that by effectively managing the symptom burden that is specifically related to physical health, there is a potential for improving symptom control in cancer patients during the COVID-19 pandemic. Our finding further reports a positive association between self-care self-efficacy and overall QoL and its domains in cancer patients supported by the integrative review of White et al. [16]. The authors concluded that higher self-care self-efficacy beliefs could improve the ability of cancer patients to perform self-care behaviors, turning to enhance their overall QoL. During the COVID-19 pandemic, when individuals face various restrictions, uncertainties, and disruptions, a strong sense of self-efficacy may help them maintain a sense of control over their own lives. This sense of control contributes to a better QoL by reducing feelings of helplessness and promoting proactive behaviors to protect one’s well-being.

In this study, the results of multiple linear regression revealed a significant association between self-care self-efficacy and all domains of QoL, as well as the overall QoL. These results partially align with an earlier study, reporting that self-care self-efficacy was only significantly associated with the physical domain of the QoL in cancer patients [45]. Another study highlighted the role of self-efficacy in enhancing QoL across various types of cancer [46]. During COVID-19, patients with cancer who had a higher level of confidence in their self-care abilities tended to better manage their symptoms after chemotherapy. Those patients who were happier and experienced less anxiety reported better psychological health. Those with strong social relationships were more proactive in maintaining their social connections through message applications and video calls. Patients with higher confidence in their ability to adjust daily routine activities and modify living environments in response to the COVID-19 pandemic were better equipped to navigate these environmental challenges. Furthermore, our findings indicate that self-care self-efficacy emerged as the primary predictor influencing every domain of QoL and the overall QoL. These suggest that individuals feel more capable of taking care of themselves and managing their circumstances, which positively affects their satisfaction and well-being despite the challenges posed by cancer and the pandemic. This implies that higher self-efficacy tends to have higher performing self-care during the COVID-19 pandemic.

### 4.1. Practice Implications

This study’s findings underscore the necessity for a holistic and tailored approach to cancer care that extends beyond medical treatment to address psychological, social, and environmental aspects of patient well-being. Key implications include enhancing self-care self-efficacy through patient education, providing targeted support for vulnerable demographics, ensuring financial and insurance assistance, and adapting treatment plans to mitigate adverse effects on QoL. Additionally, fostering community and peer support networks, along with the regular monitoring of cancer patients with psychological distress and social isolation, can significantly improve their overall QoL, especially in the context of challenges posed by the COVID-19 pandemic. These insights highlight the importance of a comprehensive, patient-centered care model that supports the multifaceted needs of cancer patients, aiming to maintain and improve their QoL during and beyond crisis situations.

### 4.2. Strengths and Limitations

The strength of this study lies in using validated instruments, including the ESAS, SUPPH, and WHOQOL BREF-THAI, which have been utilized in various cancer studies within Thailand. These instruments were applied in this study to ensure that the findings are culturally relevant and directly applicable to Thai cancer patients, thereby enhancing the credibility of the study’s outcomes. Additionally, all participants in this study were diagnosed and received chemotherapy during the COVID-19 pandemic. The study ensures that its findings are specifically relevant and can be generalized to the population of cancer patients who were diagnosed and treated during this period. This approach enhances the applicability and relevance of the study results to a specific context, which is crucial for understanding the impact of the pandemic on cancer care and patient outcomes. This study acknowledges limitations due to the small sample size of the reference group used for comparison. As a result, the findings regarding the QoL of cancer patients receiving chemotherapy, before and during the COVID-19 pandemic, may lack generalizability to the broader population of patients with cancer.

## 5. Conclusions

Cancer patients receiving chemotherapy during the COVID-19 pandemic reported a moderate level of QoL across all domains, as well as in their overall QoL. Notably, higher levels of education and self-care self-efficacy were associated with improvements in all domains of QoL and the overall QoL. Furthermore, predictive factors, including self-care self-efficacy, marital status, the burden of cancer symptoms, health insurance, and chemotherapy sequencing, accounted for 49.40% of the variance in the overall QoL. This study also indicated the crucial role of self-care self-efficacy in enhancing all domains of QoL for cancer patients undergoing chemotherapy. Therefore, interventions should prioritize the enhancement of cancer patients’ self-care self-efficacy following chemotherapy, and these efforts should be seamlessly integrated into their overall care plans. Furthermore, conducting a study into the QoL of cancer patients undergoing chemotherapy during the COVID-19 pandemic, particularly in the context of the healthcare system, was of paramount importance. This study serves as a cornerstone for the optimization of patient care, resource allocation, and informed policy decisions. It ensures that healthcare services remain adaptable and responsive to the evolving needs of these patients, thereby paving the way for enhanced outcomes and elevated QoL.

## Figures and Tables

**Table 1 ijerph-21-00317-t001:** Distribution of baseline characteristics and ESAS and SUPPH scores (N = 415).

Variables	*n* (%)
**Age** (years), *M* ± *SD*	56 ± 11
**Sex**	
Male	102 (24.6)
Female	313 (75.4)
**Marital status**	
Single	90 (21.7)
Married	252 (60.7)
Divorced/widowed	73 (17.6)
**Education level**	
Primary school	155 (37.3)
Secondary school	153 (36.9)
University or higher	103 (24.8)
**Occupation**	
Retire/housewives	163 (39.3)
Government employees	36 (8.7)
Farmer	21 (5.1)
Personal business	72 (17.3)
Labor	123 (29.6)
**Sufficient income**	
Yes	185 (44.6)
No	230 (55.4)
**Health insurance**	
Universal health coverage	243 (58.55)
Social security	91 (21.93)
Government/state enterprise employee	69 (16.63)
Private insurance	12 (2.89)
**Cancer type**	
Head and neck cancers	29 (7.0)
Lung cancer	29 (7.0)
Breast cancer	142 (34.2)
Gastrointestinal cancer	121 (29.2)
Gynecological cancer	94 (22.7)
**Stage of the disease**	
I	22 (5.3)
II	84 (20.2)
III	143 (34.5)
IV	166 (40.0)
**Sequence of chemotherapy**	
<4 cycles	235 (56.63)
4–6 cycles	69 (16.63)
>6 cycles	111 (26.74)
**ESAS score,** *M* ± *SD*	
Physical domain	8.57 ± 1.72
Psychological domain	3.59 ± 0.9
Total score	12.16 ± 2.34
**SUPPH score,** *M* ± *SD*	
Stress reduction	35.86 ± 8.21
Decision-making	10.48 ± 2.57
Positive attitude	56.80 ± 12.0
Total score	103.14 ± 21.71

Note. Cancer symptom burden is depicted by the ESAS score. Self-care self-efficacy is depicted by the SUPPH score. Abbreviations: ESAS, Edmonton Symptom Assessment System; SUPPS, Strategies Used by People to Promote Health; and SD, Standard deviation.

**Table 2 ijerph-21-00317-t002:** Comparison between the mean (SD) scores of WHOQOL-BREF-THAI in cancer survivors receiving chemotherapy in the previous study ^a^ as a control cohort (N = 192) and the current cohort (N = 415) of this study.

Domains	Mean (SD)	Levels of QoL	*t*	*p*	Mean Difference	95% CI for Mean Difference	Hedges’ g
						Lower Upper	
Physical health							
Control cohort	22 (2.59)	Moderate	−3.07	0.002	−0.84	−1.37 −0.30	−1.16
Current cohort	22.84 (3.35)	Moderate					
Psychological health							
Control cohort	16.54 (2.2)	Moderate	−16.80	<0.001	−5.09	−5.68 −4.49	−1.37
Current cohort	21.63 (3.92)	Moderate					
Social relationships							
Control cohort	11.54 (2.17)	Good	9.04	<0.001	1.73	1.35 2.11	0.79
Current cohort	9.81 (2.2)	Moderate					
Environment							
Control cohort	32.24 (5.03)	Good	9.44	<0.001	4.49	3.56 5.42	0.81
Current cohort	27.75 (5.63)	Moderate					
Overall QoL							
Control cohort	90.38 (8.04)	Moderate	2.72	0.007	3.24	0.9 5.58	0.11
Current cohort	87.14 (15.57)	Moderate					

Note. ^a^ WHOQOL-BREF-THAI of Mechalearn et al. [29].

**Table 3 ijerph-21-00317-t003:** Factors related QoL scores (WHOQOL-BREF-THAI) according to the baseline characteristics (N = 415).

Variables	Physical Health	Psychological Health	Social Relationships	Environment	Overall
**Age** ^a^	r= −0.125 *	r= −0.047	r= −0.155 **	r = −0.018	r = −0.072
**Sex** ^b^, *M* ± *SD*					
Male	22.98 ± 3.23	21.71 ± 3.53	9.93 ± 2.03	27.92 ± 5.42	87.69 ± 14.79
Female	22.80 ± 3.40	21.61 ± 4.04	9.77 ± 2.25	27.70 ± 5.71	86.96 ± 15.83
	*t* = 0.48, *p* = 0.183	*t* = 0.22, *p* = 0.825	*t* = 0.63, *p* = 0.529	*t* = 0.35, *p* = 0.730	t = 0.41, *p* = 0.682
**Marital status** ^c^, *M* ± *SD*					
Single	22.58 ± 3.57	20.93 ± 4.50	9.60 ± 2.63	26.58 ± 6.51	83.74 ± 18.29
Married	23.17 ± 3.27	21.98 ± 3.53	10.03 ± 2.05	28.37 ± 5.22	89.12 ± 14.19
Divorced/widowed	22.03 ± 3.30	21.30 ± 4.33	9.33 ± 2.02	27.08 ± 5.62	84.49 ± 15.59
	*F* = 3.72, *p* = 0.025 *	*F* = 2.68, *p* = 0.070	*F* = 3.40, *p* = 0.033 *	*F* = 4.04, *p* = 0.018 *	F = 5.33, *p* = 0.005 *
**Education level** ^c^, *M* ± *SD*					
Primary school	22.32 ± 3.38	21.31 ± 3.86	9.34 ± 2.27	27.20 ± 5.74	85.58 ± 16.17
Secondary school	22.88 ± 3.17	21.39 ± 3.95	9.82 ± 2.15	27.28 ± 5.75	85.88 ± 15.72
University/higher	23.54 ± 3.53	22.54 ± 3.92	10.54 ± 1.99	29.26 ± 5.14	91.47 ± 13.93
	*F* = 4.20, *p* = 0.016 *	*F* = 3.62, *p* = 0.028 *	*F* = 9.57, *p < 0*.001 **	*F* = 5.02, *p < 0*.007 *	*F* = 5.321, *p < 0*.005 *
**Occupation** ^c^, *M* ± *SD*					
Retile/Housewife	22.36 ± 3.26	21.04 ± 3.88	9.60 ± 2.08	27.53 ± 5.61	85.10 ± 14.99
Government employees	23.86 ± 3.85	23.08 ± 3.95	10.69 ± 2.32	30.14 ± 6.08	94.06 ± 17.04
Farmer	22.38 ± 2.89	21.19 ± 3.34	9.19 ± 1.91	27.90 ± 5.94	87.05 ± 15.66
Personal business	22.81 ± 3.54	21.67 ± 3.54	9.56 ± 1.96	27.32 ± 5.24	86.71 ± 13.95
Labor	23.28 ± 3.20	22.04 ± 4.15	10.10 ± 2.41	27.58 ± 5.64	88.08 ± 16.22
	*F* = 2.33, *p* = 0.055	*F* = 2.60, *p* = 0.063	*F* = 3.09, *p* = 0.061	*F* = 1.83, *p* = 0.122	*F* = 2.65, *p* = 0.053
**Sufficient income** ^b^, *M* ± *SD*					
Yes	23.44 ± 3.51	22.08 ± 4.08	10.09 ± 2.46	28.30 ± 5.84	88.71 ± 16.86
No	22.37 ± 3.15	21.27 ± 3.76	9.59 ± 1.94	27.32 ± 5.43	85.87 ± 14.36
	*t*= −3.24, *p* = 0.001*	*t*= −2.09, *p* = 0.038 *	*t*= −2.23, *p* = 0.026 *	*t*= −1.77, *p* = 0.078	*t*= −1.86, *p* = 0.064
**Health insurance** ^c^, *M* ± *SD*					
Universal health coverage	22.52 ± 3.46	21.36 ± 3.91	9.51 ± 2.12	27.35 ± 5.71	85.57 ± 15.48
Social security	23.26 ± 2.99	21.85 ± 3.71	10.12 ± 2.31	27.41 ± 5.12	87.66 ± 14.64
Government enterprise employee	23.55 ± 3.40	22.61 ± 3.99	10.46 ± 2.13	29.65 ± 5.75	92.23 ± 16.16
Private insurance	22.08 ± 2.84	19.83 ± 4.49	9.92 ± 2.50	27.67 ± 5.60	85.67 ± 16.22
	*F* = 2.47, *p* = 0.061	*F* = 2.78, *p* = 0.054	*F* = 4.29, *p* = 0.005 *	*F* = 3.20, *p* = 0.023 *	*F* = 3.42, *p* = 0.017 *
**Cancer type** ^c^, *M* ± *SD*					
Head and neck cancers	22.59 ± 2.96	21.41 ± 3.42	9.72 ± 2.03	26.93 ± 6.36	86.31 ± 15.04
Lung cancer	22.83 ± 3.37	20.69 ± 3.41	9.62 ± 2.11	27.31 ± 4.61	84.69 ± 13.18
Breast cancer	22.42 ± 3.59	21.32 ± 4.51	9.88 ± 2.50	27.14 ± 6.43	85.61 ± 17.50
Gastrointestinal cancer	23.26 ± 3.17	22.12 ± 3.32	9.79 ± 1.95	28.50 ± 4.74	89.26 ± 14.04
Gynecological cancer	23.03 ± 3.30	21.84 ± 3.94	9.83 ± 2.13	28.11 ± 5.42	87.71 ± 15.10
	*F* = 1.17, *p* = 0.322	*F* = 1.20, *p* = 0.310	*F* = 0.12, *p* = 0.980	*F* = 1.25, *p* = 0.288	*F* = 1.14, *p* = 0.338
**Stage of the disease** ^c^, *M* ± *SD*					
I	21.32 ± 3.51	19.86 ± 4.97	9.27 ± 2.33	25.95 ± 7.29	79.73 ± 20.29
II	23.02 ± 3.22	21.90 ± 4.13	10.31 ± 2.15	28.42 ± 5.54	89.06 ± 14.70
III	23.17 ± 3.25	21.80 ± 3.62	9.66 ± 2.14	27.30 ± 5.36	86.80 ± 14.95
IV	22.67 ± 3.45	21.58 ± 3.88	9.76 ± 2.24	28.05 ± 5.64	87.44 ± 15.66
	*F* = 2.20, *p* = 0.087	*F* = 1.74, *p* = 0.158	*F* = 2.14, *p* = 0.095	*F* = 1.60, *p* = 0.188	*F* = 2.15, *p* = 0.093
**Sequence of chemotherapy** ^c^, *M* ± *SD*					
<4 cycles	22.37 ± 3.36	21.11 ± 3.93	9.76 ± 2.18	27.03 ± 5.73	85.08 ± 15.59
4–6 cycles	23.35 ± 3.10	22.55 ± 3.51	9.68 ± 2.05	29.03 ± 4.80	90.04 ± 12.90
>6 cycles	23.53 ± 2.36	22.17 ± 3.98	10 ± 2.32	28.50 ± 5.80	89.69 ± 16.47
	*F =* 5.58, *p = 0*.004 *	*F =* 5.16, *p = 0*.006 *	*F* = 0.59, *p* = 0.556	*F* = 4.80, *p* = 0.009 *	F = 4.85, *p* = 0.008 *
**ESAS** ^a^	r = −0.112 *	r = −0.073	r = −0.023	r = −0.001	r = −0.039
**SUPPH** ^a^	r = 0.502 **	r = 0.588 **	r = 0.636 **	r = 0.630 **	r = 0.684 **

Note. ^a^ Pearson’s correlation; ^b^ Independent t-test; and ^c^ One-way ANOVA. Cancer symptom burden is depicted by the ESAS score. Self-care self-efficacy is depicted by the SUPPH score. * *p* < 0.05.; ** *p* < 0.001. Abbreviations: SUPPS, Strategies used by People to Promote Health; ESAS, Edmonton Symptom Assessment System; and SD, Standard deviation.

**Table 4 ijerph-21-00317-t004:** Stepwise multiple linear regression analysis of factors associated with QoL (N = 415).

QoL Domains	Variables	B	SE	*ꞵ*	t	*p*	R^2^ Change	Model Adjusted R^2^
Physical health								30.20%
	**SUPPS**	0.076	0.006	0.489	11.75	<0.001	0.252	
	**ESAS**	−0.228	0.060	−0.159	−3.83	<0.001	0.022	
	**Marital status** (ref, single)							
	Divorced/widowed	−1.079	0.363	−0.123	−2.972	0.003	0.014	
	**Sequence of chemotherapy** (ref, <4 cycles)							
	>6 cycles	−0.766	0.279	−0.113	−2.74	0.006	0.013	
	**Sufficient income** (ref, No)							
	Yes	0.652	0.279	0.097	2.34	0.020	0.009	
Psychological health								36.90%
	**SUPPS**	0.106	0.007	0.586	14.87	<0.001	0.346	
	**ESAS**	−0.205	0.066	−0.122	−3.122	0.002	0.013	
	**Sequence of chemotherapy** (ref, <4 cycles)							
	>6 cycles	−0.760	0.311	−0.096	−2.45	0.015	0.009	
	**Health insurance** (ref, universal health coverage)							
	Private insurance	−1.874	0.913	−0.080	−2.05	0.041	0.006	
Social relationships								41.60%
	**SUPPS**	0.063	0.004	0.618	16.29	<0.001	0.404	
	**Health insurance** (ref, universal health coverage)							
	Government enterprise employee	0.601	0.223	0.102	2.69	0.007	0.008	
	**Age**	−0.018	0.008	−0.091	−2.39	0.017	0.006	
Environment								41%
	**SUPPS**	0.161	0.010	0.619	16.36	<0.001	0.397	
	**Health insurance** (ref, universal health coverage)							
	Government enterprise employee	1.681	0.572	0.111	2.94	0.003	0.012	
	**Sequence of chemotherapy** (ref, <4 cycles)							
	4–6 cycles	1.145	0.571	0.076	2.01	0.046	0.006	
Overall								49.40%
	**SUPPS**	0.481	0.025	0.670	18.93	<0.001	0.468	
	**Marital status** (ref, single)							
	Married	3.020	1.128	0.095	2.68	0.008	0.011	
	**ESAS**	−0.597	0.235	−0.090	−2.54	0.011	0.009	
	**Health insurance** (ref, universal health coverage)							
	Government enterprise employee	3.511	1.476	0.084	2.38	0.018	0.007	
	**Sequence of chemotherapy** (ref, <4 cycles)							
	>6 cycles	−2.203	1.108	−0.070	−1.99	0.047	0.005	

Note. Cancer symptom burden is depicted by the ESAS score. Self-care self-efficacy is depicted by the SUPPH score. Abbreviations: SUPPS, Strategies used by People to Promote Health; ESAS, Edmonton Symptom Assessment System.

## Data Availability

Data availability for this study is subject to reasonable requests made to the corresponding author.

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
