# Peer review of "Quality of Life and Associated Factors among Cancer Patients Receiving Chemotherapy during the COVID-19 Pandemic in Thailand"

_ijerph, 2024, doi:10.3390/ijerph21030317_

Round 1
Reviewer 1 Report
Comments and Suggestions for Authors
Cancers, cancer patient's experience and quality of care is very important issues and critical in connection with international health policy.
Introduction:
Line 30-34: At the beginning of the introduction, you can add some details about cancer indicators - in your country and in the world.
There is a lack of definition of the aim of the study.
Materials and methods:
In the section materials and methods, you don’t need capital letters.
Were the data anonymised since you were obtaining consent for the study?
In section Measures – please explain what do you extracted informations from the patient's medical records.
Edmonton Symptom Assessment Scale (ESAS)-Cancer symptom burden is well prepared.
Line 164-166: The internal consistency of the total SUPPH in this study 164 was assessed using Cronbach's α coefficient, resulting in a value of 0.88, which indicates 165 a high level of internal consistency and reliability – please explain.
The tools you used in the study are well described, but missing from the material and methods section is a summary of how many questionnaires you used and why each was adapted to the population and how you adapted it.
Results:
Table 3 is too large and unclear. You should maybe cut them into a few parts?
Practical implications is very well.
Conclusion:
All conclusions is justified and supported by the results.
Author Response
The authors thank the reviewer for the thoughtful reviews and insightful review of our manuscript. Your comments and suggestions were very helpful in improving the overall quality of the paper, and we are grateful for the time and effort you put into providing such valuable feedback. We have addressed the comments point-by-point below.
Introduction section:
Comments 1: Line 30-34: At the beginning of the introduction, you can add some details about cancer indicators - in your country and in the world. Response 1: We agree with this comment. Therefore, we haveadded some details of cancer statistics in the world and Thailand- page 2, paragraph 1, lines 34-38.
Comments 2: There is a lack of definition of the aim of the study. Response 2: We agree with this comment. Therefore, we have revised the aims of this study- page 3, paragraph 4, lines 101-103.
Materials and methods section:
Comments 3: In the section materials and methods, you don’t need capital letters.
Response 3: We agree with this comment. Therefore, we have revised it to be “Materials and Methods”- page 4, lines 105.
Comments 4: Were the data anonymised since you were obtaining consent for the study? Response 4: Yes, the data wasanonymized by returning it in sealed envelopes. Therefore, we have added this sentence - page 5, paragraph 1 of the procedure section, lines 145-146.
Comments 5: In section Measures – please explain what do you extracted informations from the patient's medical records.
Response 5: Data were gathered from patients’ medical records after obtaining authorization from the National Cancer Institute to access these documents. Therefore, we have added more details to this - page 6, paragraph 2 of the measures section, lines 162-165.
Comments 6: Line 164-166:
The internal consistency of the total SUPPH in this study was assessed using Cronbach's α coefficient, resulting in a value of 0.88, which indicates a high level of internal consistency and reliability – please explain.
Response 6: We agree with this comment. Therefore, we have revised this sentence that the internal consistency of the total SUPPH in this study was assessed using Cronbach's α coefficient, resulting in a value of 0.88, which indicates higher than acceptable internal consistency- page 7, paragraph 4 of the measures section, lines 192-194.
Comments 7: The tools you used in the study are well described, but missing from the material and methods section is a summary of how many questionnaires you used and why each was adapted to the population and how you adapted it.
Response 7: We agree with this comment. Therefore, we have added some explanation as a suggestion- page 6, paragraph 1 of the measuressection, lines 153-159.
Results section:
Comments 8: Table 3 is too large and unclear. You should maybe cut them into a few parts?
Response 8: We agree with this comment. Therefore, Table 3 has been revised by reducing the number of columns and organizing the space more efficiently. However, the variables presented in that table are related to the same topic; therefore, we have reported them together. We also have revised Table 1 to be consistent with Table 3 - pages 22-25.
Reviewer 2 Report
Comments and Suggestions for Authors
Dear authors,
please see the attached file with my specific comments to your manuscript

Author Response
The authors thank the reviewer for the thoughtful reviews and insightful review of our manuscript. Your comments and suggestions were very helpful in improving the overall quality of the paper, and we are grateful for the time and effort you put into providing such valuable feedback. We have addressed the comments point-by-point below.
Introduction section:
Comments 1: Lines 30-31: Please specify whether this is a worldwide estimate. Response 1: We agree with this comment. Therefore, we have revised it to be “Cancer is a prominent global contributor to mortality, accounting for nearly one in every six deaths, based on worldwide estimates”- page 2, paragraph 1, lines 33-34.
Comments 2: Lines 37-38: “Healthcare professionals…due to healthcare need”. The last part of the sentence does not make sense. Please rephase. Response 2: We agree with this comment. Therefore, we have revised it to be “ Healthcare professionals are under pressure to continue to ensure the clinical needs of patients with cancer are still met during this time”- page 2, paragraph 1, lines 42-43.
Comments 3: Lines 48-50: “It is imperative…in Thailand” I think this is conclusive statement must be moved to the end of the introduction section. Response 3: We agree with this comment. Therefore, we have deleted this sentence because it is not consistent with the sentences at the end of the introduction section.
Comments 4: Lines 51-69: The content of the paragraph need to be reorganized. Authors must first report the impact of cancer and chemotherapy in different aspects of QoL and then report the specific impact of the pandemic in cancer treatment and the subsequent QoL. Additionally, authors must provide a definition of self-efficacy (line 64) since it is not obvious that all readers are aware of it. If authors have assumed in advance the role of self-efficacy, perhaps they must consider its inclusion in the title. Response 4: We agree with this comment. Therefore, we have detailed the impact of chemotherapy on cancer patients’ quality of life (QoL) and then described its effects during the COVID-19 pandemic- page 2, paragraph 2, lines 53-55. Additionally, we have elaborated on self-efficacy and self-care self-efficacy in the context of cancer- page 3, paragraph 2, lines 70-78. Given the numerous factors related to QoL identified in this study, we have maintained the original title.
Comments 5: Line 70: Authors must specify what constitutes of “the unique cultural and social context of Thailand”. Similarly, (lines 73-75) in which ways of management differ between Thai patients and those in other regions? Response 5: We agree with this comment. Therefore, we have detailed the unique cultural and social context of Thailand, including the role of Buddhist philosophy- page 3, paragraph 3, lines 83-87. Additionally, we have described how management practices differ between Thai patients and those in other regions, focusing on the healthcare system and traditional Thai health beliefs and practices- pages 3-4, paragraph 3, lines 89-97.
Materials and Methods section
Comments 6: Line 114: “The research…?” Please use a unique term, study or research all over the text. Response 6: We agree with this comment. Therefore, we have replaced the word “the research” with “study” throughout the entire manuscript-page 5 with line 138, page 11 with line 335, and page 17 with lines 495, 497, and 510.
Comments 7: Line 160: “The author…” Which author? Response 7: We agree with this comment. The “author” of this study is a person who translated the SUPPH questionnaire. Therefore, we have revised this sentence- pages 7, paragraph 4 of the measure section, lines 189-192.
Comments 8: Lines 191-193: Authors could provide Cronbach’s alpha for the WHOQOL subdomains. Response 8: We agree with this comment. Therefore, we have provided the score of Cronbach’s α for the WHOQOL subdomains. “The Cronbach’s α coefficients were found to be 0.83 for physical health, 0.87 for psychological health, 0.84 for social relationships, 0.85 for the environment, and 0.85 for the total scale”- pages 8, paragraph 5 of the measures section, lines 217-220.
Results section
Comments 9: I am not sure whether reporting the possible scores in Table 1 adds something extra. There are mentioned anyway at the description of the tools (page 4). I suggest to remove these scores from the Table. Response 9: We agree with this comment. Therefore, we have deleted the possible score presented in Table 1.
Comments 10: Line 265: “QoL scores among the age of the participants…”. This stage does not make sense. Please rephrase. Response 10: We agree with this comment. Therefore, we have revised it to be “The age of the participants was negatively associated with the domains of physical health (r = -0.125; p < 0.05) and social relationships (r = -0.155; p < 0.001)”- pages 10, paragraph 3, lines 294-295.
Comments 11: Lines 278-283 & lines 312-315. These sentences look like a discussion rather than a presentation of results. Authors are advised to move and use them at the discussion section. Response 11: We agree with this comment. Therefore, these sentences have been deleted.
Discussion section
Comments 12: Line 329: “In Thailand, the public health system has implemented new healthcare system…”. It looks like a tautology. Please correct. Response 12: We agree with this comment. Therefore, we have deleted those sentences and provided the details of the healthcare system's adaptation by using the home care services- pages 12, paragraph 1, lines 341-345.
Comments 13: Line 341: “The possible reason for this study…”. Do you mean “…for this finding”? Response 13: We agree with this comment. Therefore, we have replaced the words “possible reason” with “…for this finding” - pages 12, paragraph 1, lines 354-356.
Comments 13: Lines 348-448: While previous discussion is rich and extensive, the discussion of the finding regarding the predictors of QoL is rather limited. Authors must discuss at a greater details the role of self-efficacy, since this factor affects all domains of QoL. Response 13: We agree with this comment. Therefore, we have provided more discussion about the role of self-efficacy in all domains of QoL- pages 15, paragraph 7, lines 442-458.
Comments 14: Lines 463-469: Authors restrict the strengths of their study in rather common feature: the comparison of QoL before and after COVID-19 and the use of a widely used QoL measure. Authors must report other strengths of their study. Response 14: We agree with this comment. Therefore, we have deleted the previously mentioned strengths of the study and provided additional strengths for this study - pages 16, paragraph 9, lines 471-479.